# Genome-Wide Association Study of Osteoporosis Risk in Korean Pre-Menopausal Women: The Korean Genome and Epidemiology Study

**DOI:** 10.3390/ijms26178177

**Published:** 2025-08-22

**Authors:** Su Kang Kim, Seoung-Jin Hong, Gyutae Kim, Ju Yeon Ban, Sang Wook Kang

**Affiliations:** 1Department of Biomedical Laboratory Science, Catholic Kwandong University, Gangneung 25601, Republic of Korea; skkim7@cku.ac.kr; 2Department of Prosthodontics, College of Dentistry, Kyung Hee University, Seoul 02447, Republic of Korea; ssabock@khu.ac.kr; 3Department of Oral and Maxillofacial Radiology, College of Dentistry, Kyung Hee University, Seoul 02447, Republic of Korea; latinum.omfr@khu.ac.kr; 4Department of Dental Pharmacology, College of Dentistry, Dankook University, Cheonan 31116, Republic of Korea; jyban@dankook.ac.kr; 5Department of Oral and Maxillofacial Pathology, College of Dentistry, Kyung Hee University, Seoul 02447, Republic of Korea

**Keywords:** osteoporosis, GWAS, polymorphism, menopause, bone mineral density

## Abstract

Osteoporosis is a common disease characterized by a reduction in bone mineral density (BMD), leading to an increased risk of pathological fractures and even mortality. Although menopause is a major risk factor, osteoporosis can also occur in premenopausal women. The aim of this study was to identify genetic variants associated with the development of osteoporosis in Korean premenopausal women. Subjects were recruited from the Anseong and Ansan cohorts of the Korean Genome and Epidemiology Study (KoGES). Clinical and epidemiological characteristics were assessed, and participants were classified based on BMD values measured at the distal radius and mid-shaft tibia. Individuals with confounding risk factors such as low body weight, smoking, high alcohol consumption, steroid/hormone therapy, or relevant medical history were excluded. A total of 247 healthy controls and 57 osteoporosis patients were included. Genotyping was performed using the Illumina Infinium HumanExome BeadChip and the Affymetrix Axiom Exome Array. Data were analyzed using the SNP and Variation Suite and PLINK, with quality control thresholds set at MAF ≥ 0.05 and HWE *p* ≥ 0.01. Functional annotation and protein structure predictions were performed using PolyPhen-2, SIFT, and PROVEAN. Genome-wide association analyses identified 113 single-nucleotide polymorphisms (SNPs) in 69 genes significantly associated with osteoporosis (*p* < 0.05) in both platforms, with 18 SNPs showing high cross-platform consistency (*p* < 0.01). Several of these genes were implicated in bone metabolism (e.g., *ESRRG*, *PECAM1*, *COL6A5*), vitamin D metabolism (e.g., *NADSYN1*, *EFTUD1*), skeletal muscle function (e.g., *PACSIN2*, *ESRRG*), and reproductive processes (e.g., *CPEB1*, *EFCAB6*, *ASXL3*). Notably, the *CPEB1* rs783540 SNP exhibited the strongest association (*p* < 0.001) in both analyses. Our findings suggest that genetic polymorphisms in pathways related to bone metabolism, vitamin D signaling, muscle–bone interaction, and reproductive hormone regulation may contribute to the development of osteoporosis in Korean premenopausal women. These results provide a genetic basis for early identification of at-risk individuals and warrant further functional studies to elucidate the underlying mechanisms.

## 1. Introduction

Osteoporosis is one of the most common age-related diseases and is characterized by a reduced bone mineral density (BMD). It is the most common cause of fractures in the elderly, and it can also lead to serious complications and even death [1]. According to the ‘Sixth Korea National Health and Nutrition Examination Survey (KNHANES VII-1), 2016, Korea Centers for Disease Control and Prevention’, osteoporosis generally occurs in the elderly (42.6% in women > 65 years), although it can also occur in younger people (5.3% in women < 65 years).

The most common causes of osteoporosis are old age and menopause. It may be caused by systemic or genetic diseases, and it is also known to be related to family history, race, nutrition, and smoking and drinking habits [2]. Previous studies have reported that body weight, nutrition, and the genotypes of *VDR* and *ER* genes affect BMD in young women [3]. Hyperparathyroidism [4], hyperthyroidism [5], excessive drinking [6], glucocorticoids [7], and so on can also lead to osteoporosis in young women. Genetic factors, such as familial history, also play a role in the value of BMD in young women [8]. There are many studies exploring the genetic causes of low BMD osteoporosis in postmenopausal women, and many genetic polymorphisms have been identified in individuals with different ethnicities. Single-nucleotide polymorphisms (SNPs) of *VDR* and *OPG* genes were found to be associated with osteoporosis in Chinese postmenopausal women [9]. SNP of *ESR1* gene has also been found to reduce BMD in postmenopausal women of southern Slovakia [10]. The polymorphism of the *RANKL* gene related to bone metabolism is also associated with osteoporosis in postmenopausal women [11]. However, there are few studies evaluating the association of genetic causes and osteoporosis in premenopausal women.

The Illumina Infinium HumanExome BeadChip targets approximately 240,000 coding variants, allowing for intensive detection of missense and nonsense variants that are critical for protein function. However, its coverage is limited to coding regions, and it exhibits relatively low statistical power for detecting rare or low-frequency variants [12]. The Affymetrix Axiom Exome Array demonstrates a very high positive predictive value (PPV) for most variants, offering high accuracy and reproducibility. However, it has a limitation: the PPV for heterozygous calls tends to be lower for extremely rare variants with a minor allele frequency (MAF) of less than 0.01% [13]. The advantages of these two technologies can complement each other’s weaknesses. As they have different design strategies and signal readout mechanisms, they could complement platform-specific technical biases and variant calling errors when used together.

Therefore, in this study, we conducted a large-scale genetic analysis using data from both the Illumina Infinium HumanExome BeadChip version 1.1 and the Affymetrix Axiom Exome Array5.0, based on participants from the Anseong and Ansan cohorts of the Korean Genome and Epidemiology Study (KoGES). The aim of this study was to identify genetic variants associated with the development of osteoporosis in Korean premenopausal women.

## 2. Results

The demographic characteristics of the subjects participating in the study are shown in Table 1.

A total of 304 subjects (57 osteoporosis patients and 247 healthy controls) were included in the analysis. Age, alcohol consumption, and calcium consumption did not significantly differ between the osteoporosis patients and healthy controls. There was no statistically significant difference between the two groups because the subjects who had a medical history of fracture or arthritis, a smoking habit, long-term steroid intake, or hormone therapy were excluded. Although previous studies have demonstrated that weight and BMI could be risk factors of osteoporosis in young women [14,15], these parameters were significantly higher in the osteoporosis patient group in this study. The BMD values, distal radius speed of sound (DR-SOS), distal radius T-score (DR-T), distal radius Z-score (DR-Z), mid-shaft tibia speed of sound (MT-SOS), mid-shaft tibia T-score (MT-T), and mid-shaft tibia Z-score (MT-Z) were significantly higher in the control group.

Figure 1 is a Manhattan plot and Q-Q plot showing the results of the SNP analysis in the Illumina Infinium HumanExome BeadChip and Affymetrix Axiom exome arrays. 1253 SNPs in the Illumina Infinium HumanExome BeadChip and 15,874 SNPs in the Affymetrix Axiom exome array were found to be statistically significant with a *p* value of <0.05.

Panel A presents the analysis results from the Illumina Infinium HumanExome BeadChip, while Panel B shows the results from the Affymetrix Axiom Exome Array. In the Manhattan plot of Panel A, significant signals are generally well distributed across chromosomes, with notable peaks in −log10(P) values observed on certain chromosomes (e.g., chromosomes 6 and 11). The Q-Q plot reveals that most SNPs lie close to the diagonal line representing the expected and observed −log10(P) values, indicating that the statistical tests were performed appropriately. However, a number of points in the upper right corner deviate from this line, suggesting the presence of SNPs with higher-than-expected significance. Similarly, the Manhattan plot in Panel B demonstrates a widespread distribution of significant signals, with some SNPs reaching −log10(P) values as high as 5.8. The corresponding Q-Q plot also aligns closely with the diagonal under the null hypothesis, though several SNPs display clear deviations, further supporting the presence of statistically significant variants.

Figure 2 shows the chromosomal locations of the statistically significant SNPs in both the Illumina Infinium HumanExome BeadChip and the Affymetrix Axiom exome array. As shown in Figure 1, 113 SNPs were found to be significant in 69 genes (*p* < 0.05). Among them, 41 SNPs with no information of genes were excluded, and the locations of the 72 SNPs are plotted in Figure 3. The red-marked locations are SNPs with a *p* value < 0.01 in both analyses.

Figure 3 shows the protein–protein interaction (PPI) network of genes that were commonly significant (*p*-value < 0.05) in the Illumina BeadChip and Affymetrix Axiom exome arrays generated using the STRING database. Six clusters were distinguished through K-means clustering (K = 6). Genes within each of these clusters are likely to share similar biological functions. Among these genes, *NCAM1*, *PECAM1*, *GNL3*, and *PTPRD* appear to be hub genes with multiple interactions within the network.

The prediction of protein damage using PolyPhen-2 version 2.2.2, SIFT 4G, and PROVEAN version 1.1 revealed that 6 SNPs (rs1799852 SNP in *TF*, rs11917356 SNP in *COL6A5*, rs2276360 in *NADSYN1*, rs1128431 SNP in *EFTUD1*, and rs7232237 and rs2282632 SNPs in *ASXL3*) were associated with damage of the protein structure. Table 2 shows the SNPs that could affect the structure of the proteins. Most of them were predicted to be benign; however, the rs1128431 SNP in the *EFTUD1* gene was predicted to have possible functional damage according to both PolyPhen-2 and SIFT.

Table 3 shows the list of statistically significant SNPs associated with osteoporosis in premenopausal women in both analyses (*p* < 0.01). A total of 18 SNPs were found to be statistically significant in both the analyses: rs783540 SNP in *CPEB1*, rs3731646 SNP in *SH3BP4*, rs10506525 SNP in *MSRB3*, rs2110871 SNP in *MAGI2*, rs2172802 SNP in *LPHN3*, rs6895902 SNP in *MAML1*, rs2020945 SNP in *PWP2*, rs3756987 SNP in *RSPH3*, rs2286550 SNP in *CATSPERG*, rs4729759 SNP in *CUX1*, rs10513680 SNP in *SAMD7*, rs1052053 SNP in *PMF1*-*BFLAP*, rs2764020 SNP in *STARD13*, rs7088318 SNP in *PIP4K2A*, rs151719 SNP in *HLA*-*DMB*, rs2302234 SNP in *FAM20A*, rs16990991 SNP in *EFCAB6*, and rs12757165 SNP in *ESRRG* gene. In particular, rs783540 exhibited the strongest association with osteoporosis, showing a *p*-value of 0.000 in both analyses. This cross-platform consistency may strengthen the reliability of these SNPs as potential candidate markers.

## 3. Discussion

Previous studies have shown the close relationship between menopause and the development of osteoporosis. Estrogen deficiency due to menopause accelerates the induction of M-CSF, RANKL, and TNF-α, which promote blood calcium concentration and bone resorption [16]. Estrogen deficiency also affects the production of parathyroid hormone and results in decreased intestinal calcium absorption and renal calcium conservation [17]. Parathyroid hormone is associated with vitamin D, which also regulates calcium absorption in the intestine [18,19]. Increased blood calcium level and decreased intestinal calcium absorption lead to decreased bone formation [20]. Estrogen is directly related to bone metabolism. Estrogen promotes the apoptosis of osteoclasts [21], and its deficiency increases osteoclastogenesis [22] as well as the apoptosis of osteocytes and osteoblasts [23].

Our results showed that many genes are associated with osteoporosis in premenopausal women. In premenopausal women, BMD could be reduced even before menopause. This implies that a mechanism other than menopause might play a role in the development of osteoporosis. Although not all the genes in our results were found to be associated with bone metabolism, several potential genes were identified.

In our results, a number of genes were found to be significantly associated with BMD value or bone formation. *ESRRG* gene encodes a member of the estrogen receptor-related receptor family. The polymorphism of *ESRRG* gene is associated with the determination of bone density in European women [24]. The protein encoded by the gene is a sex- and RUNX2-dependent negative regulator of postnatal bone formation, as demonstrated in mice [25]. *NTN4* gene encodes netrin-4, which promotes the differentiation and migration of osteoblasts and inhibits the differentiation of osteoclast [26,27]. *TF* gene encodes transferrin, which is related to BMD [28]. The polymorphism of this gene is associated with an increased risk of osteonecrosis of the femoral head [29]. The *CLEC* gene is closely related to adaptive immunity. The polymorphism of the *CLEC16A* gene could cause an alteration in the leukocyte count and lead to reduced BMD and fractures in elderly women [30]. In our results, the genes encoding adhesion molecules were also found to be involved in bone formation and resorption. *PECAM1* gene encodes the platelet and endothelial cell adhesion molecule 1, and the protein encoded by this gene is a negative regulator of monocyte-derived osteoclastogenesis. Loss of this gene increases osteoclastogenesis and leads to bone loss [31]. *NCAM1* gene encodes a cell adhesion molecule in the immunoglobulin superfamily, and it plays various roles in cell differentiation, including osteogenesis [32]. The roles of *GGT1* and *COL6A5* genes in bone metabolism are well studied in rats but less so in humans. The function of *GGT1* gene in humans is not well understood, but mutation of *GGT1* gene in rats has been found to promote the development of osteoclasts and increase bone resorption, resulting in osteoporosis [33]. *GGT1* is an activator of TLR4-mediated osteoclastogenesis, and *GGT1* is overexpressed in transgenic mice exhibiting symptoms of osteoporosis [34]. *COL6A5* is present in almost all the tissues in mice, but only in the skin, lung, testis, colon, and small bowel in humans. Therefore, studies in humans generally focus on the skin and the small intestine [34]. However, another study reported that the polymorphism in *COL6A5* gene could be linked to variations in BMD in both mouse and human [35]. *STARD13* gene encodes a protein that may be involved in the regulation of cytoskeletal reorganization, cell proliferation, and so on. Some recent microRNA-based studies have shown that miR-125 is up-regulated in patients with osteoporosis, and *STARD13* is the target of it [36,37].

It is well known that vitamin D plays an important role in BMD and the development of osteoporosis. Our results demonstrated the association between SNPs in several genes associated with vitamin D metabolism and the development of osteoporosis in premenopausal women. *NADSYN1* gene encodes NAD synthetase 1, which plays an important role in the synthesis of nicotinamide adenine dinucleotide. The polymorphism of *NADSYN1* gene is associated with vitamin D levels and metabolic profile [38]. *NADSYN1* gene is one of the vitamin D pathway genes, the polymorphism of which affects the level of vitamin D in pregnant women [39]. *EFTUD1* gene is a target gene for vitamin D in the mammary cells [40], and it also plays an important role in mediating the pro-apoptotic effects of vitamin D in breast cancer [41]. Table 2 shows that the rs1128431 SNP of *EFTUD1* gene can have a severe effect on the protein structure. This means that the rs1129431 SNP can affect the function of *EFTUD1* protein.

The relationship between muscle and osteoporosis is well known. Decreased muscle strength is related to decreased BMD [42], and a decline in the function of skeletal muscle results in osteoporosis [43]. It is reported that the *PACSIN2* and *ESRRG* genes, which were found to be significant in our results, are associated with skeletal muscle exercise. Differential changes in the expression were observed in transcriptome analysis during aerobic exercise [44].

The genes associated with the reproductive system were found to be significantly associated with osteoporosis in our results. Androgen deficiency is a common cause for the development of osteoporosis, and androgen is used for the treatment of osteoporosis [45]. The *EFCAB6* gene was called *DJBP* and is mainly expressed in the testis. The protein encoded by this gene plays a role in the negative regulation of the androgen receptor [46]. *ASXL3* is mostly expressed in the bone [47]. It is reported that mutations in *ASXL3* gene are associated with sporadic primary hyperparathyroidism and recurrently mutated in sporadic parathyroid adenomas [48]. *ASXL3* is a gene associated with the androgen pathway and is regulated by androgen in the human neural cells [49]. *ESRRG* gene takes part in one of the estrogen pathways and may affect estrogenic response related to females [50]. *NTN4* encodes netrin-4, which is required for the development of the mammary gland and is expressed in the normal breast epithelium. It plays an important role in the prognosis of ER-positive breast cancer [51]. As mentioned above, *EFTUD1* is also associated with breast cancer. *CPEB1* gene is another interesting gene, and the rs783540 SNP of *CPEB1* gene has shown very high significance in both analyses in our study (*p* < 0.001). Some previous studies have shown the relationship between *CPEB1* and the reproductive system. A previous microRNA study reported that the *CPEB1* gene is a target of miR-3646 and is downregulated in breast and ovarian cancers [52]. *CPEB1* gene plays an important role in oocyte maturation. It regulates mRNA translation during oocyte maturation [53]. The polymorphism of *CPEB1* gene might result in premature ovarian insufficiency [54]. A recent study has reported that oocyte maturation is associated with BMD, and superovulation decreases BMD [55]. CATSPERG is reported to be an auxiliary subunit of the CATSPER calcium channel complex, primarily expressed in the principal piece of the sperm flagellum. Recent bioinformatic analyses have identified CATSPERG as a hub gene significantly associated with spermatogenesis in non-obstructive azoospermia [56]. This might suggest potential roles beyond male reproduction that merit further investigation in the context of bone metabolism.

Several genes were studied to be associated with the development of osteoporosis in premenopausal women. It is well known that menopause and aging are the major causes of osteoporosis. However, other causes might also be involved in the development of osteoporosis, as it can also develop in the younger women. Other causes of osteoporosis include smoking, drinking, a history of fracture, steroid and hormone therapy, systemic diseases, genetic factors, and so on. Therefore, the other factors were matched in this study as much as possible, and the subjects who were young and premenopausal women were involved in this analysis.

Table 4 summarizes the significant genes found to be associated with the development of osteoporosis in premenopausal women in this study. The genes that have been reported to be associated with osteogenesis, osteoclastogenesis, and BMD value were shown to be associated with osteoporosis in premenopausal women. Additionally, the genes involved in vitamin D metabolism, the vitamin D pathway, and skeletal muscle exercise were shown to be significantly associated with osteoporosis. Finally, the genes associated with the reproductive process were significantly associated with osteoporosis. It is well known that menopause and estrogen deficiency are associated with the development of osteoporosis. It is important to note that the polymorphisms of genes involved in the reproductive processes or sex hormones are also associated with the development of osteoporosis in premenopausal women.

In this study, we identified various biological pathways associated with osteoporosis based on SNPs that were significantly detected using both platforms. Genetic pathways related to estrogen signaling and the reproductive system were observed in some variants, suggesting a potential association between hormonal changes and bone density loss in premenopausal women. Additionally, genes involved in ovarian function, germ cell maturation, androgen receptor regulation, and post-transcriptional regulation were also implicated in bone density regulation, further supporting the influence of hormonal mechanisms on early bone loss. Moreover, gene functions related to osteogenesis, differentiation of osteoblasts and osteoclasts, microRNA regulation, extracellular matrix organization, vitamin D metabolism, and muscle–bone interactions were identified. These findings indicate that osteoporosis is likely influenced by complex interactions among hormonal, metabolic, immune, and inter-tissue signaling pathways, rather than by a single molecular mechanism.

The main limitation of this study is the relatively small sample size. However, the incidence of osteoporosis in premenopausal women is low, and when excluding individuals with known risk factors such as alcohol use, steroid medication, or fracture history, the rate becomes even lower [57]. In this study, we sought to minimize the influence of external risk factors in order to more clearly evaluate the role of genetic predisposition. Additionally, the use of two independent genotyping platforms applied to the same population enhanced the reliability of our findings. The consistently detected variants may play a significant role in the development of osteoporosis in young Korean women and could serve as foundational data for future functional studies and the development of predictive genetic models.

## 4. Materials and Methods

### 4.1. Study Subjects

The subjects analyzed in this study were obtained from the Korean Genome and Epidemiology Study cohort. A total of 7077 subjects from three phases of the Anseong and Ansan Cohort Study were screened, and finally 304 subjects (57 osteoporosis patients and 247 healthy controls) were selected. Figure 4 shows the process of selection of 304 subjects from a pool of 7077 subjects. First, men and postmenopausal women were excluded. Subsequently, T-scores and Z-scores were analyzed. Since the study focused exclusively on premenopausal women, the Z-score was also included as a criterion for assessment. Thus, the subjects with a T-score < −2.5 and a Z-score < −2.0 were classified as osteoporosis patients, and the subjects with a T-score > −1.0 were classified as controls. The subjects with −2.5 < T-score < −1.0 were excluded from the analysis. To analyze the influence of the genetic risk factors on osteoporosis, other risk factors were controlled, such as environmental factors, general factors, habits, medical history, and so on [12]. The subjects with a body weight < 58 kg, body mass index (BMI) < 19, alcohol consumption > 10 mL/day, smoking history, medication history of long-term steroid or hormone therapy, thyroid disorder, fracture, or arthritis were excluded. There were no data on parathyroid disorder or vitamin D consumption. As few subjects consumed calcium supplements more than the recommended dosage, they were excluded, and the subjects who consumed less than the recommended dosage of calcium supplements were included in the study. Finally, 57 osteoporosis patients and 247 healthy controls were included in the study. The demographic characteristics of the subjects are shown in Table 1. This study was approved by the Institutional Review Board of Dankook University (IRB no. 2018-08-004).

### 4.2. Statistical Analysis

To compare the osteoporosis patients and healthy controls, an independent Student’s t-test was performed using PAWS 18.0 (SPSS, Chicago, IL, USA). The results of 30,538 SNPs for each subject were obtained using the Illumina Infinium HumanExome BeadChip, and the results of 242,901 SNPs for each subject were obtained using an Affymetrix Axiom exome array (Affymetrix, Santa Clara, CA, USA). A logistic regression analysis was performed to examine the association between the development of osteoporosis and genetic polymorphisms using SNP and Variation Suite (Golden Helix, Bozeman, MT, USA) and PLINK software, version 1.9 [58]. SNPs with MAF ≥ 0.05 and HWE *p*-value ≥ 0.01 were removed. Manhattan plots and QQ plots were plotted using the SNPEVG program, version 3.1 [59]. PhenoGram was used for the scheme of the location of the SNPs in chromosomes (http://visualization.ritchielab.org (accessed on 9 January 2019)). PolyPhen-2 version 2.2.2, SIFT 4G, and PROVEAN version 1.1 were used for predicting protein damage by the SNPs [60,61,62].

## 5. Conclusions

Our results showed that the genes associated with bone metabolism, vitamin D metabolism, skeletal muscle exercise, and reproductive processes were closely associated with the development of osteoporosis in premenopausal women. Among these, the *CPEB1* gene, which was associated with oocyte maturation, showed a strong and consistent association in both analyses. Our results showed that the development of osteoporosis in premenopausal women might have a relationship with genetic factors involved in various mechanisms. Further studies on these genes should be conducted in the future.

## Figures and Tables

**Figure 1 ijms-26-08177-f001:**
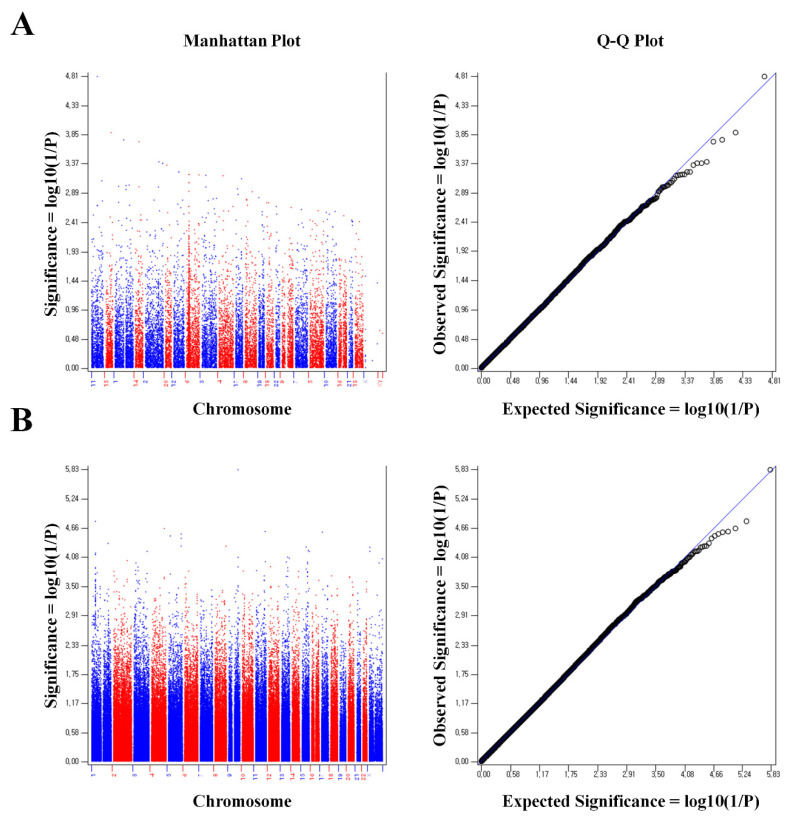
Manhattan plot and Q-Q plot of GWAS. (**A**): Illumina Infinium HumanExome BeadChip. (**B**): Affymetrix Axiom exome array (*p* value of the threshold line = 0.01).

**Figure 2 ijms-26-08177-f002:**
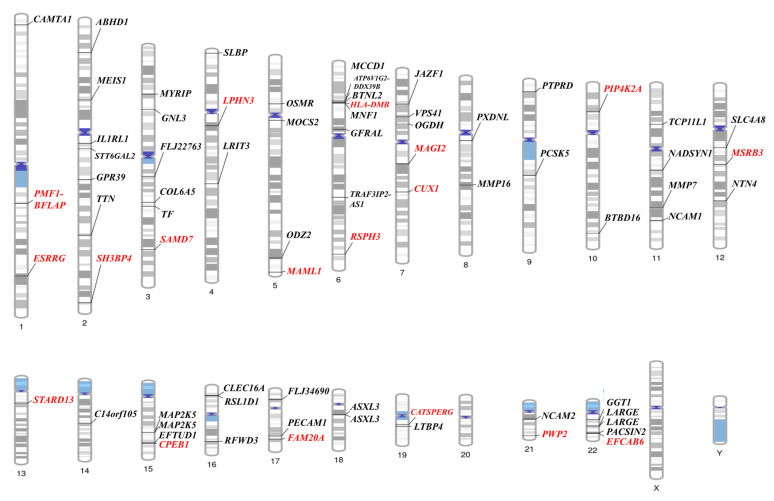
Significant gene areas where linkage with osteoporosis in the premenopausal women are represented using a phenogram (*p* < 0.05 in both the Illumina Infinium HumanExome BeadChip and the Affymetrix Axiom exome array. Red; *p* < 0.01). Gene names may appear more than once if multiple significant SNPs are located within the same gene.

**Figure 3 ijms-26-08177-f003:**
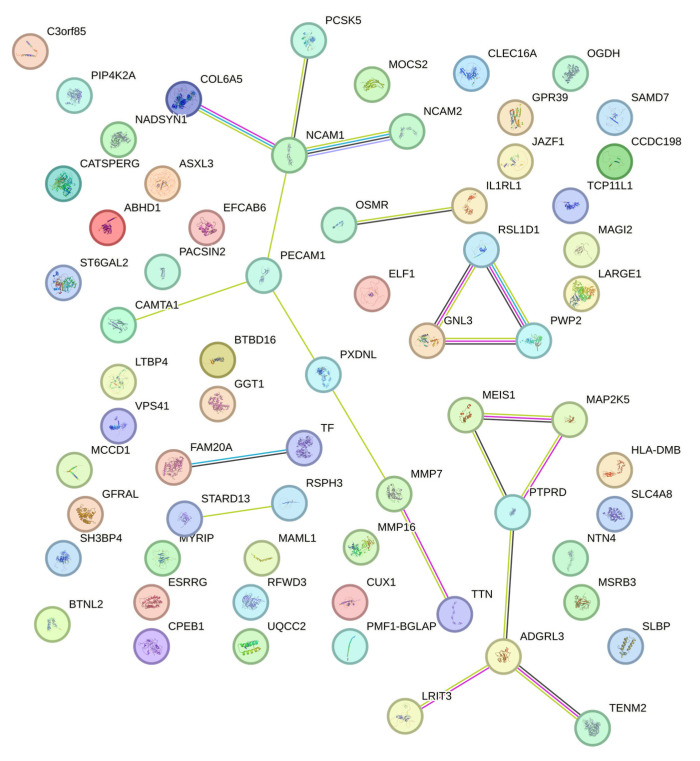
Protein–Protein Interaction Network.

**Figure 4 ijms-26-08177-f004:**
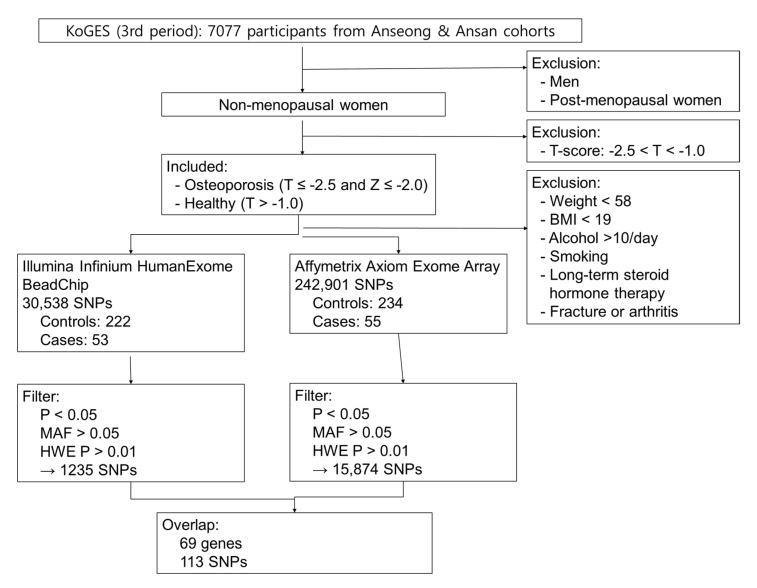
Flow chart of selection of the study subjects.

**Table 1 ijms-26-08177-t001:** Demographic characteristics of study subjects with *p* values for group comparisons. Data are presented as mean ± standard deviation.

	Control (n = 247)	Osteoporosis (n = 57)	*p* Value
Age (years)	47.08 ± 2.57	47.54 ± 2.46	0.218
Weight (kg)	63.94 ± 5.13	66.62 ± 7.98	0.018
BMI (kg/m^2^)	25.71 ± 2.35	27.23 ± 3.22	0.001
Alcohol consumption (g/day)	1.06 ± 2.03	0.66 ± 1.35	0.072
Calcium consumption (mg/day)	435.88 ± 196.07	485.01 ± 205.45	0.092
Medical history of fracture	none	none	
Medical history of arthritis	none	none	
Smoking	none	none	
Long-term steroid	none	none	
Hormone therapy	none	none	
DR-SOS (m/s)	4269.38 ± 123.94	4107.46 ± 152.21	0.000 *
DR-T (m/s)	0.8 ± 0.99	−0.45 ± 1.19	0.000 *
DR-Z (m/s)	0.92 ± 1.02	−0.29 ± 1.22	0.000 *
MT-SOS (m/s)	3959.12 ± 65.93	3608.74 ± 90.86	0.000 *
MT-T (m/s)	0.001 ± 0.63	−3.4 ± 0.89	0.000 *
MT-Z (m/s)	0.2 ± 0.63	−3.2 ± 0.92	0.000 *

BMI: body mass index; DR: distal radius; SOS: speed of sound; MT: mid-shaft tibia. * *p* < 1.0 × 10^−12^.

**Table 2 ijms-26-08177-t002:** SNPs that could affect the structure of proteins.

SNP	Chromosome	Position	Reference Allele	Alternate Allele	Gene	Amino Acid Change	PolyPhen-2		SIFT		PROVEAN	
							Score	Prediction	Score	Prediction	Score	Prediction
rs1799852	3	133475722	C	T	*TF*	p.Leu247=	-	-	0.333	tolerated	0.00	neutral
rs11917356	3	130110550	A	G	*COL6A5*	p.Asp982Val	0.093	benign	0.717	tolerated	−1.79	neutral
rs2276360	11	71169547	G	C	*NADSYN1*	p.Val74Leu	0.000	benign	1.000	tolerated	2.56	neutral
rs1128431	15	82456227	T	C	*EFTUD1*	p.Ile617Val	0.791	possibly damaging	0.010	deleterious	−1.00	neutral
rs7232237	18	31324934	A	G	*ASXL3*	p.Met1708Val	0.000	benign	0.668	tolerated	−0.84	neutral
rs2282632	18	31320229	A	G	*ASXL3*	p.Asn954Ser	0.003	benign	0.744	tolerated	−0.73	neutral

**Table 3 ijms-26-08177-t003:** A list of the overlapped SNPs associated with osteoporosis between the statistically significant SNPs in the Illumina Infinium HumanExome BeadChip (*p* < 0.01) and the Affymetrix Axiom array (*p* < 0.01) in the Korean premenopausal women (logistic regression analysis).

SNP	Gene	Chromosome	Position	*p* Value (Exome)	*p* Value (Affymetrix)
rs783540	*CPEB1*	15	83254708	0.000	0.000
rs3731646	*SH3BP4*	2	235950002	0.000	0.003
rs10506525	*MSRB3*	12	65783378	0.001	0.003
rs2110871	*MAGI2*	7	78080548	0.002	0.002
rs2172802	*LPHN3*	4	62453209	0.003	0.001
rs6895902	*MAML1*	5	179201847	0.004	0.001
rs2020945	*PWP2*	21	45528919	0.004	0.003
rs3756987	*RSPH3*	6	159398700	0.004	0.010
rs2286550	*CATSPERG*	19	38861362	0.004	0.008
rs4729759	*CUX1*	7	101536886	0.005	0.004
rs10513680	*SAMD7*	3	169644710	0.005	0.000
rs1052053	*PMF1-BFLAP*	1	156202173	0.005	0.008
rs2764020	*STARD13*	13	34234642	0.006	0.003
rs7088318	*PIP4K2A*	10	22852948	0.007	0.001
rs151719	*HLA-DMB*	6	32903900	0.007	0.005
rs2302234	*FAM20A*	17	66538239	0.007	0.008
rs16990991	*EFCAB6*	22	44167684	0.008	0.003
rs12757165	*ESRRG*	1	216716537	0.009	0.003

**Table 4 ijms-26-08177-t004:** List of the significant genes with possible mechanisms of the development of osteoporosis (*p* < 0.05).

SNP	Chr.	Position	Gene	FC	*p* Value (Exome)	*p* Value (Affy)	Possible Mechanism in Osteoporosis	Function	Refs.
rs12757165	1	216716537	*ESRRG*	INT	0.009	0.003	Bone mineral density	Determination of bone density	[24]
rs1799852	3	133475722	*TF*	SYN	0.029	0.005	Osteoclastogenesis	Bone mineral density	[28]
rs1436109	11	112991618	*NCAM1*	INT	0.012	0.001	Osteogenesis	Osteogenesis	[32]
rs4341610	12	96149288	*NTN4*	INT	0.029	0.027		To promote osteoblasts and inhibit osteoclast	[26,27]
rs6498142	16	11081249	*CLECL16A*	INT	0.046	0.030		Bone mineral density	[30]
rs11917356	3	130110550	*COL6A5*	MIS	0.014	0.005		Variation in bone mineral density	[35]
rs2812	17	62401118	*PECAM1*	3′ UTR	0.016	0.027		Negative regulator of Osteoclastogenesis	[31]
rs4820599	22	24990213	*GGT1*	INT	0.003	0.041		Osteoclastogenesis	[33]
rs2764020	13	34234642	*STARD13*	INT	0.006	0.003		Target of miR-125, which is up-regulated in Osteoporosis	[36]
rs2276360	11	71169547	*NADSYN1*	MIS	0.038	0.027	Vitamin D	Vitamin D status and metabolic profile	[38,39]
rs1128431	15	82456227	*EFTUD1*	MIS	0.025	0.032		Target gene for vitamin D	[36,40]
rs12757165	1	216716537	*ESRRG*	INT	0.009	0.003	Skeletal muscle	Skeletal muscle exercise	[44]
rs11090122	22	43308475	*PACSIN2*	INT	0.045	0.028		Skeletal muscle exercise	[44]
rs12757165	1	216716537	*ESRRG*	INT	0.009	0.003	Reproductive system	Estrogen pathways	[24,50]
rs16990991	22	44167684	*EFCAB6*	INT	0.008	0.003		Regulation of androgen receptor	[46]
rs4341610	12	96149288	*NTN4*	INT	0.029	0.027		Prognosis of ER-positive breast cancer	[51]
rs783540	15	83254708	*CPEB1*	INT	0.000	0.000		Oocyte maturation	[53,54]
rs7232237	18	31324934	*ASXL3*	MIS	0.015	0.011		Androgen pathway	[49]
rs2282632	18	31320229	*ASXL3*	MIS	0.019	0.038		Androgen pathway	[49]
rs1128431	15	82456227	*EFTUD1*	MIS	0.025	0.032		Breast cancer	[41]
rs2286550	19	38861362	*CATSPERG*	MIS	0.004	0.008		Spermatogenesis	[56]

SNP: single-nucleotide polymorphism. Chr.: chromosome, Refs.: references, FC: functional consequence, INT: intron, SYN: synonymous, MIS: missense, 3′ UTR: 3′ untranslated region, Exome: Illumina Infinium HumanExome BeadChip, Affy: Affymetrix Axiom Exome Array.

## Data Availability

The data presented in this study are not publicly available due to ethical restrictions. The dataset contains personally identifiable information (e.g., genetic information or patient data) that could compromise the privacy of the participants. However, the results from the statistical analyses, such as the regression analysis, can be made available upon reasonable request from the corresponding author to ensure the reproducibility of the study’s findings.

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
