# Peer review of "Genome-Wide Association Study of Osteoporosis Risk in Korean Pre-Menopausal Women: The Korean Genome and Epidemiology Study"

_ijms, 2025, doi:10.3390/ijms26178177_

Round 1
Reviewer 1 Report
Comments and Suggestions for Authors
Dear Authors
The topic fits in the journal's scope.
The question study is original and well-defined. The study design is appropriate and understandable. However, the authors decided to put the 4.Materials and Methods section, after the results, and subsection numeration starts at 2.1?
The Abstract, need to be improved: methods is missing the statistical analysis; wide conclusion and no implications.
Also the aim of the study is diferent from the one in the abstract.
the title of table 1, is not according with the data, because also have a p value and is missing the n sample of each group.
Table 3 need to be according the format of the publication.
Table 4 needs to be presented in a different format, easier for the reader.
At the conclusion, it is not recommended to use this type of expression."..had the most statistically significant association in both the analyses (P < 0.001)." .
There are implications that remain to be explored beyond further studies.
Regards
Author Response
We sincerely thank you for taking the time to review our manuscript and for providing valuable and constructive feedback. Please find our detailed, point-by-point responses below. The corresponding revisions and corrections have been incorporated into the manuscript and are clearly indicated using highlights in the resubmitted files for your convenience.
Comments 1: the authors decided to put the 4.Materials and Methods section, after the results, and subsection numeration starts at 2.1?
Response 1: Thank you for pointing this out. We agree with this comment. Therefore, We have changed ‘2.1. Study subjects’ and ‘2.2. Statistical analysis’ to ‘4.1. Study subjects’ and ‘4.2. Statistical analysis’.
Comments 2: The Abstract, need to be improved: methods is missing the statistical analysis; wide conclusion and no implications.
Response 2: Thank you for pointing this out. We agree with this comment. Therefore, We have changed whole abstract to ‘Background: Osteoporosis is a common disease characterized by a reduction in bone mineral density (BMD), leading to an increased risk of pathological fractures and even mortality. Although menopause is a major risk factor, osteoporosis can also occur in premenopausal women. The aim of this study was to identify genetic variants associated with the development of osteoporosis in Korean premenopausal women.
Methods: Subjects were recruited from the Anseong and Ansan cohorts of the Korean Genome and Epidemiology Study (KoGES). Clinical and epidemiological characteristics were assessed, and participants were classified based on BMD values measured at the distal radius and mid-shaft tibia. Individuals with confounding risk factors such as low body weight, smoking, high alcohol consumption, steroid/hormone therapy, or relevant medical history were excluded. A total of 247 healthy controls and 57 osteoporosis patients were included. Genotyping was performed using the Illumina Infinium HumanExome BeadChip and the Affymetrix Axiom Exome Array. Data were analyzed using the SNP & Variation Suite and PLINK, with quality control thresholds set at MAF ≥ 0.05 and HWE P ≥ 0.01. Functional annotation and protein structure predictions were performed using PolyPhen-2, SIFT, and PROVEAN.
Results: Genome-wide association analyses identified 113 single nucleotide polymorphisms (SNPs) in 69 genes significantly associated with osteoporosis (P < 0.05) in both platforms, with 18 SNPs showing high cross-platform consistency (P < 0.01). Several of these genes were implicated in bone metabolism (e.g., ESRRG, PECAM1, COL6A5), vitamin D metabolism (e.g., NADSYN1, EFTUD1), skeletal muscle function (e.g., PACSIN2, ESRRG), and reproductive processes (e.g., CPEB1, EFCAB6, ASXL3). Notably, the CPEB1 rs783540 SNP exhibited the strongest association (P < 0.001) in both analyses.
Conclusions: Our findings suggest that genetic polymorphisms in pathways related to bone metabolism, vitamin D signaling, muscle–bone interaction, and reproductive hormone regulation may contribute to the development of osteoporosis in Korean premenopausal women. These results provide a genetic basis for early identification of at-risk individuals and warrant further functional studies to elucidate the underlying mechanisms.” to add the statistical analysis, results, conclusion and implications.
Comments 3: the aim of the study is diferent from the one in the abstract.
Response 3: Agree. We have, accordingly, changed the sentence “The purpose of this study was to investigate the genetic pathogenesis of osteoporosis in Korean premenopausal women.” to “The aim of this study was to identify genetic variants associated with the development of osteoporosis in Korean premenopausal women.”
Comments 4: the title of table 1, is not according with the data, because also have a p value and is missing the n sample of each group.
Response 4: Agree. We have, accordingly, changed the title of table 1 to ‘Demographic characteristics of study subjects with P values for group comparisons. Data are presented as mean ± standard deviation.’. And sizes of each group (‘n=247’, ‘n=57’) were added in table 1.
Comments 5: Table 3 need to be according the format of the publication.
Response 5: Agree. Thank you for your feedback. I have modified the format of the table as per your suggestion.
Comments 6: Table 4 needs to be presented in a different format, easier for the reader.
Response 6: We agree with your comment and thank you for this helpful suggestion. Therefore, we have reformatted Table 4 in a horizontal (landscape) layout to improve readability and have abbreviated certain terms to make the table more concise. (e.g., SNP: single-nucleotide polymorphism. Chr.: chromosome, Ref.: reference, FC: functional consequence, INT: intron, SYN: synonymous, MIS: missense, 3’ UTR: 3′ un-translated region, Exome: Illumina Infinium HumanExome BeadChip, Affy: Affymetrix Axiom Exome Array.)
Comments 7: At the conclusion, it is not recommended to use this type of expression."..had the most statistically significant association in both the analyses (P < 0.001)." .
Response 7: Agree. Thank you for your comment. I've revised the sentence to avoid overemphasizing or misinterpreting the statistical results, as you pointed out. The sentence was changed to ‘showed a strong and consistent association in both analyses’.

Reviewer 2 Report
Comments and Suggestions for Authors
In general the paper is important for Osteoporosis society and might be accepted after improvement.
I have several questions and comments:
Questions and comments
- In Table 1 there are no units for alcohol consumption, calcium consumption and speed of sound listed. Also, there is no designation, what type of data are presented in the table, is it mean ± standard deviation?
- On Figure 1A there are no significant signals from X-chromosome, and on Figure 1B the significant signal intensity from X-chromosome is similar to the other chromosomes. Could the authors clarify this difference between two analysis results?
- What is shown in blue and cyan on Figure 2? One can assume that blue color denotes centromeric regions, but the meaning of cyan color is not clear.
- On Figure 2 some genes (MAP2K5, LARGE and ASXL3) are listed twice. It’s unclear whether this means the occurrence of two different SNPs in this genes or gene duplications, since the figure caption starts with “Significant gene areas”.
- The meaning of an asterisk near the CPEB1 gene on Figure 2 is unclear. Also, it’s better to use italics for gene names on this Figure.
- In Table 2 authors list SNPs that could affect the structure of proteins, but for better understanding it’s necessary to describe the amino acid change resulting from these SNP’s.
- Authors do not discuss the role of the statistically significant SNP in CATSPERG According to Uniprot, CATSPERG protein is the cation channel sperm-associated auxiliary subunit gamma, is expressed in testis and localized in the sperm principal piece. But this significant SNP was found in pre-menopausal women, and its association with osteoporosis remains unclear.
Comments on the Quality of English Language
English must be improved
Author Response
We sincerely thank you for taking the time to review our manuscript and for providing valuable and constructive feedback. Please find our detailed, point-by-point responses below. The corresponding revisions and corrections have been incorporated into the manuscript and are clearly indicated using highlights in the resubmitted files for your convenience
Comments 1: In Table 1 there are no units for alcohol consumption, calcium consumption and speed of sound listed. Also, there is no designation, what type of data are presented in the table, is it mean ± standard deviation?
Response 1: Thank you for pointing this out. We agree with this comment. Therefore, we have added the sentence ‘Data are presented as mean ± standard deviation.’ in the title of table 1. And we have add the unit for alcohol consumption (g/day), calcium consumption (mg/day) and speed of sound (m/sec).
Comments 2: On Figure 1A there are no significant signals from X-chromosome, and on Figure 1B the significant signal intensity from X-chromosome is similar to the other chromosomes. Could the authors clarify this difference between two analysis results?
Response 2: Thank you for your valuable comment. The Illumina Infinium HumanExome BeadChip used in Figure 1A contains X-chromosome variants. However, during the quality control process (minor allele frequency, and Hardy–Weinberg equilibrium filtering), a large proportion of X-chromosome SNPs did not pass the thresholds in our dataset of premenopausal women and were therefore excluded from the final analysis, resulting in no significant X-chromosome signals in Figure 1A.
Comments 3: What is shown in blue and cyan on Figure 2? One can assume that blue color denotes centromeric regions, but the meaning of cyan color is not clear.
Response 3: Figure 2 was generated using Phenogram (https://visualization.ritchielab.org/). The blue banding color is from PhenoGram cytoband styling and not related to data significance.
Comments 4: On Figure 2 some genes (MAP2K5, LARGE and ASXL3) are listed twice. It’s unclear whether this means the occurrence of two different SNPs in this genes or gene duplications, since the figure caption starts with “Significant gene areas”.
Response 4: Agree. We appreciate the reviewer’s comment. The repeated appearance of the same gene name (e.g., MAP2K5, LARGE, ASXL3) in Figure 2 indicates the presence of more than one significant SNP within that gene. We have clarified this in the figure caption to avoid confusion. The sentence ‘Gene names may appear more than once if multiple significant SNPs are located within the same gene.’ is added to the title of figure 2.
Comments 5: The meaning of an asterisk near the CPEB1 gene on Figure 2 is unclear. Also, it’s better to use italics for gene names on this Figure.
Response 5: Agree. We appreciate your comment. In the original figure, the asterisk indicated that the SNP in CPEB1 had a p-value of less than 0.001 in both the Illumina Infinium HumanExome BeadChip and the Affymetrix Axiom Exome Array. However, as this information was already described later in the manuscript and the meaning in the figure was unclear, we have removed the asterisk in the revised version. In addition, all gene names in the figure have been reformatted in italics for consistency.
Comments 6: In Table 2 authors list SNPs that could affect the structure of proteins, but for better understanding it’s necessary to describe the amino acid change resulting from these SNP’s.
Response 6: Agree. In accordance with the comment, we have revised Table 2 to include an additional column describing the amino acid change resulting from each SNP, based on the MANE Select transcript for the corresponding gene. This addition should facilitate a better understanding of the potential structural and functional impact of the listed SNPs. Column ‘Amino Acid Change’, amino acid changes ‘p.Leu247=’, ‘p.Asp982Val’, ‘p.Val74Leu’, ‘p.Ile617Val’, ‘p.Met1708Val’, and ‘p.Asn954Ser’ were added in the table 2.
Comments 7: Authors do not discuss the role of the statistically significant SNP in CATSPERG According to Uniprot, CATSPERG protein is the cation channel sperm-associated auxiliary subunit gamma, is expressed in testis and localized in the sperm principal piece. But this significant SNP was found in pre-menopausal women, and its association with osteoporosis remains unclear.
Response 7: Agree. We appreciate your point regarding CATSPERG's potential relevance to sperm. We found that recent bioinformatic analysis identified CATSPERG as a hub gene significantly associated with spermatogenesis in non-obstructive azoospermia (NOA), suggesting a functional role in male germ cell development. While its direct involvement in bone metabolism remains speculative, this evidence strengthens the biological relevance of CATSPERG and supports future investigation into pleiotropic effects that may extend beyond reproductive physiology. We added the following sentence to the discussion section and added reference; ‘CATSPERG is reported to be an auxiliary subunit of the CATSPER calcium channel com-plex, primarily expressed in the principal piece of the sperm flagellum. Recent bioinfor-matic analyses have identified CATSPERG as a hub gene significantly associated with spermatogenesis in non-obstructive azoospermia [56]. This might suggest potential roles beyond male reproduction that merit further investigation in the context of bone metabolism.’ and ‘56. Li, C.; Li, M.; Liu, Y.; Li, J.; Zhang, Y.; Wang, H.; Zhang, Y.; Jia B., Identification of Potential Biomarkers Associated with Spermatogenesis in Azoospermia. Clin Lab 2024, 70, (11).’

Round 2
Reviewer 1 Report
Comments and Suggestions for Authors
Dear authors, thank you for the opportunity to review your work.
Thank you for agreeing with all the proposed improvements and for implementing them effectively. The document has been improved in terms of writing and formatting, contributing to its readability and interpretation. The bibliographical references are adequate and support the study. They have also been reviewed. At this stage, the work meets the conditions for publication. You took care to acknowledge the limitations of the research and make them clear to the reader.
Congratulations on your work.
Reviewer 2 Report
Comments and Suggestions for Authors
Authors corrected all things I have noited out. The paper is improved and can be accepted